# The Properties of Proinflammatory Ly6C^hi^ Monocytes Are Differentially Shaped by Parasitic and Bacterial Liver Infections

**DOI:** 10.3390/cells11162539

**Published:** 2022-08-16

**Authors:** Stefan Hoenow, Karsten Yan, Jill Noll, Marie Groneberg, Christian Casar, Niels Christian Lory, Malte Vogelsang, Charlotte Hansen, Vincent Wolf, Helena Fehling, Julie Sellau, Hans-Willi Mittrücker, Hannelore Lotter

**Affiliations:** 1Department of Molecular Parasitology and Immunology, Bernhard Nocht Institute for Tropical Medicine, 20359 Hamburg, Germany; 2Institute for Immunology, University Medical Center Hamburg-Eppendorf, 20246 Hamburg, Germany; 3Bioinformatic Facility, University Medical Center Hamburg-Eppendorf, 20246 Hamburg, Germany

**Keywords:** *Entamoeba histolytica*, *Listeria monocytogenes*, liver infection, inflammatory Ly6C^hi^ monocytes, surface marker, ROS production

## Abstract

In the past, proinflammatory CD11b^+^Ly6C^hi^ monocytes were predominantly considered as a uniform population. However, recent investigations suggests that this population is far more diverse than previously thought. For example, in mouse models of *Entamoeba (E.) histolytica* and *Listeria (L.) monocytogenes* liver infections, it was shown that their absence had opposite effects. In the former model, it ameliorated parasite-dependent liver injury, whereas in the listeria model it exacerbated liver pathology. Here, we analyzed Ly6C^hi^ monocytes from the liver of both infection models at transcriptome, protein, and functional levels. Paralleled by *E. histolytica*- and *L. monocytogenes*-specific differences in recruitment-relevant chemokines, both infections induced accumulation of Ly6C^+^ monocytes at infection sites. Transcriptomic analysis revealed a high similarity between monocytes from naïve and parasite-infected mice and a clear proinflammatory phenotype of listeria-induced monocytes. This was further reflected by the upregulation of M2-related transcription factors (e.g., *Mafb, Nr4a1, Fos*) and higher CD14 expression by Ly6C^hi^ monocytes in the *E. histolytica* infection model. In contrast, monocytes from the listeria infection model expressed M1-related transcription factors (e.g., *Irf2, Mndal, Ifi204*) and showed higher expression of CD38, CD74, and CD86, as well as higher ROS production. Taken together, proinflammatory Ly6C^hi^ monocytes vary considerably depending on the causative pathogen. By using markers identified in the study, Ly6C^hi^ monocytes can be further subdivided into different populations.

## 1. Introduction

Monocytes are a type of mononuclear phagocyte that, as cells of the innate immune system, are part of the initial immune response to invading pathogens [1]. Under homeostatic conditions, monocytes patrol the blood stream, replenish macrophage pools in tissues, and are recruited rapidly to sites of infection and inflammation [2,3]. Egress from the bone marrow is mediated mainly by C-C chemokine receptor 2 (CCR2), which binds to CCL2 secreted by cells in injured or infected tissue [4,5].

Once in the tissues, these cells shape the inflammatory milieu via expression of pro- or anti-inflammatory cytokines, phagocytic activity, and antigen presentation; they can also differentiate into macrophages [3,6]. However, monocytes can trigger immunopathology when inadequately controlled [7,8]. Murine monocytes are mostly identified as CD11b^+^ Ly6C^+^ Ly6G^−^ cells and are commonly divided into two major subsets: proinflammatory CD11b^+^Ly6C^hi^ and anti-inflammatory CD11b^+^Ly6C^lo^ monocytes [9,10]. They can be further subdivided according to the expression of CCR2 and CX3CR1: Ly6C^hi^ CCR2^hi^ CX3CR1^int^ (proinflammatory) and Ly6C^lo^ CX3CR1^hi^ CCR2^-^ (anti-inflammatory), although the usage of CX3CR1 is presently questioned [11]. Fate mapping and single cell approaches revealed the priming of Ly6C^hi^ monocytes towards a neutrophil-like monocyte (nMO) or dendritic cell-like monocyte (dcMO) phenotype under steady state conditions, and to *Cxcl10*^+^ and *Saa3*^+^ monocytes in pathogenic conditions [12,13,14]. In two murine models of liver infection, the absence of Ly6C^hi^ monocytes results in opposite disease outcomes. In the murine model of hepatic amebiasis, intrahepatic infection with the protozoan parasite *E. histolytica* results (as in humans) in focal liver destruction (18). This type of liver damage, also termed amebic liver abscess (ALA), is almost abolished in mice with a *Ccr2* knockout (*Ccr2*^-/-^) or in mice in which monocyte were immunodepleted, suggesting an immunopathologic role for inflammatory monocytes (14). This liver damage also depends on inflammatory factors and can be inhibited by a specific blockade of TNF-α or by general immunosuppression [15,16]. By contrast, in the murine model of *L. monocytogenes* infection, monocytes play a protective role, as indicated by the higher bacterial load and increased granuloma formation in the liver of *Ccr2*^-/-^ mice (18).

The aim of this study was to characterize liver Ly6C^hi^ monocyte subsets from both infection models with regard to their histological localization within infected liver areas, their transcriptome and surface marker expression, and their production of reactive oxygen species (ROS). This study revealed functionally distinct inflammatory Ly6C^hi^ monocytes in these two infection models and identified a novel combination of a transcription factor and surface markers that allow to distinguish proinflammatory monocyte subsets during inflammatory states.

## 2. Materials and Methods

### 2.1. Mice

All murine studies complied with relevant ethical regulations for animal testing and research. Animal experiments were performed in accordance with the German animal protection laws and were reviewed by the federal health authorities of the State of Hamburg in accordance with the ARRIVE guidelines (N082/2018). C57BL/6J (WT) and *Ccr2*^-/-^ [17] were bred and kept in individually ventilated cages under specific pathogen-free conditions at the animal facility at the Bernhard Nocht Institute for Tropical Medicine, *Cd38*^-/-^ [18] mice were kept in the facility of the University Medical Center Hamburg-Eppendorf. The mice were kept with a day/night cycle of 12 h, humidity of 50–60%, and a temperature of 21 °C. Mice were euthanized with CO_2_ with a replacement rate of 20–30% of the cage volume per minute, followed by cervical dislocation and cardiac puncture.

### 2.2. Infection of Mice with E. histolytica and L. monocytogenes

Male C57BL/6 mice (aged 8–12 weeks) were used for infections. Briefly, 2 × 10^5^ trophozoites of the highly pathogenic clone B2, generated from cell line B (HM-1:IMSS), were suspended in 20 µL of incomplete TYI-S-33 medium and injected into the left liver lobe, as described previously [19]. Abscess size was calculated as % abscessed left liver lobe. Mice were infected with 2 × 10^4^
*L. monocytogenes* strain EGD or 1 × 10^7^
*L. monocytogenes* ∆*actA* in 200 µL PBS via the lateral tail vein. Bacterial inoculi were controlled by plating serial dilutions on tryptic soy broth agar plates at 37 °C.

### 2.3. Immunohistochemistry

Liver tissue from *E. histolytica-* or *L. monocytogenes*-infected mice were fixed in formalin (4%) and embedded in paraffin. Sections (0.2 µm) were stained with H&E or immunostained with antibodies specific for CD11b (EPR1344; 1:1000) and Ly6C (ER-MP20; 1:400) using standard procedures. Antibodies were detected using DCS SuperVision Single Species horseradish peroxidase (HRP)-Polymer (Innovative Diagnostic-Systems) and sections were then counterstained with hemalaun.

### 2.4. Cytokine Measurement

Liver tissue from naïve and infected mice was homogenized using a 70 µm cell strainer and 1mL of lysis buffer (0.05% Tween 20; one tablet Protease inhibitor (Roche) per 50 mL) and centrifuged (10.000× *g*, 10 min, 4 °C). The supernatant was used for cytokine analysis using a customized LegendPLEX kit (CCL2, CCL3, TNF-α, IL-1β, IFN-γ, IL-10, IL-13; BioLegend, San Diego, CA, USA).

### 2.5. Detection of Reactive Oxygen Species

CM-H2DCFDA staining reagent (Thermo Fisher, Waltham, MA, USA, C6827) was used for staining ROS. A stock solution (50 µg/100 µL Ethanol) was prepared, diluted 1:100 in DPBS (PAN Biotech, Aidenbach, Germany, P04-361000) and added to 1.5–2.0 × 10^6^ pelleted immune cells. After 30 min at 37 °C, the cells were washed twice, diluted in antibody staining solution, and measured by flow cytometry.

### 2.6. RNA Sequencing and Data Analysis

Hepatic monocytes of *E. histolytica*- or *L. monocytogenes*-infected C57BL/6 mice were sorted on the indicated days p.i., with a purity of 80–85% and surface staining was performed using CD11b (APC-Cy7; Ly6C (FITC) and Ly6G (APC). Cells were then sorted into collection tubes containing 2 mL of RNAprotect cell reagent (Qiagen). RNA was isolated using the RNeasy Plus Micro Kit (Qiagen) and RNA integrity was analyzed using an Agilent 6000 Pico Kit and an Agilent 2100 Bioanalyzer (Agilent). Samples used for transcriptome sequencing fulfilled the following criteria: total RNA ≥ 200 ng (≥20 ng/µL); RNA integrity number (RIN) ≥7.0, 28S/18S ≥ 1.0. RNA library preparation and sequencing were performed by BGI Genomics, China. The data comprised paired-end short reads. All raw data were aligned to the mouse reference genome GRCm38 Ensembl 85 and the corresponding Gencode annotation using STAR [20] (version 2.5.2a). Differential expression analysis was performed in R (version 3.3.3; R Foundation for Statistical Computing, Austria) using DESeq2 (version 1.13.8) [21]. Reads from different lanes per replicate were combined after checking for the absence of batch effects on a PCA plot. We performed differential gene expression analysis between both infection models within the Ly6C^hi^ and Ly6C^lo^ monocytes respectively using the day after infection as additional co-variate in the DESeq design. To test for differential expression across all three time points (naive, d3 p.i, d5 p.i.), we used a likelihood ratio test for each monocyte group using time point, infection model, and an interaction term for both variables as full design and compared it against the reduced model with the interaction term removed. A threshold of 0.05 for Benjamini–Hochberg corrected *p* values was used to determine significance by the “statistical overrepresentation test”. Gene set analysis was performed using PANTHER GO-slim (Version 16) [22], with *p* adj < 0.05. Heatmaps were created using http://heatmapper.ca (accessed on 1 July 2019) combining time points 3 and 5. Volcano plots were made using GraphPad Prism V8.4.3.

### 2.7. Isolation of Immune Cells and Flow Cytometry

Organs were collected immediately after euthanasia. Mouse blood was collected by cardiac puncture immediately after the mice were euthanized and collected in EDTA-coated tubes. Immune cells were obtained after performing two erythrocyte lysis steps. Hepatic immune cells were isolated using a Percoll gradient. Homogenized liver was suspended in 80% Percoll (GE Healthcare) and overlaid with 40% Percoll diluted in complete RPMI 1640 (Gibco) containing 10% fetal calf serum, L glutamine, and penicillin/streptavidin (cRPMI). After a centrifugation step without brake (800× *g*; 25 min; 21 °C), cells localized in the interphase of the two Percoll layers were transferred to a new collection tube for further washing steps with PBS and erythrocyte cell lysis. The obtained single cell suspension was washed twice with cRPMI. Spleen cells were separated with a cell strainer (70 µm), centrifuged (300× *g*) and washed with PBS before two erythrocyte lysis steps. Bone marrow-derived immune cells were collected by flushing out the bone marrow from isolated thighs with a cannula and 2 mL PBS, followed by separation of cells with a cell strainer. Monocytes from spleen and bone marrow were purified using the EasySept^TM^ Monocyte Isolation Kit (StemCell Technologies) according to the manufacturer’s instructions.

Flow cytometry analysis was performed on immune cells (2 × 10^6^) from liver (Supplementary: blood, bone marrow, spleen). Staining was performed using murine Fc-Blocking solution, followed by fixation of cells in 1% paraformaldehyde. The following antibodies were used:

CD45.1 BUV395 (A20) (Becton Dickinson); CD11b BV510 (M1/70), CD14 BV421 (1Sa14-2), CD38 PeCy7 (90), CD45.2 PeCy5 (30-F11), CD74 AF647 (In1/CD74), CD86 AF700 (GL-1), Ly6C FITC/PE (HK1-4), Ly6G BV785 (1A8) (BioLegend); IRF2 AF488 (sc-374327) and mafb PerCP (OTI1E9) (Novus Biologicals). Analysis was performed on a BD LSR Fortessa and a Cytek Aurora, and data were analyzed using FlowJo software V10.7.1.

### 2.8. Quantitative RT-PCR

RNA was isolated from liver tissue using TRIzol reagent (Life Technologies, Carlsbad, CA, USA) after chloroform extraction, isopropanol precipitation, and ethanol washing steps. RNA was transcribed into cDNA using Maxima First Strand cDNA Kit (Thermo Fisher Scientific, Waltham, MA, USA). Gene expression levels were calculated using the 2-ddct method and the following primers were used: *Ncf1*: fwd 5-AAGCTCCTGGATGGCTGGTG-3, rev 5-CCTGGCGCTCACCCTTTGT-3; *Rps9*: fwd 5-GCTAGACGAGAAGGATCCCC-3, rev 5-TTGCGGACCCTAATGTGACG-3. The annealing temperature was set at 60 °C. The qPCR was performed (in 384 well plates) with Maxima SYBR Green qPCR Master Mix (Thermo Scientific, Waltham, MA, USA) and a Roche LightCycler 480.

### 2.9. Uniform Manifold Approximation and Projection for Dimension Reduction (UMAP)

Overall, 270,000 cells (90,000 (naïve); 89,999 (*L. monocytogenes*-infected); 90,000 (*E. histolytica*-infected) Ly6C^hi^ cells, expressing CD11b, CD14, CD38, CD74, CD86, and IRF2, were used for concatenation according to McInnes et al. (McInnes, L., Healy, J. & Melville, J. UMAP: *Stat. Mach. Learn*. *arXiv* Uniform manifold approximation and projection for dimension reduction. *preprint* arXiv:1802.03426 (2018)).

### 2.10. Statistical Analysis

Statistical analysis was carried out using either a parametric paired or unpaired two-tailed Student’s t-test (normal distribution) or a nonparametric two-tailed Mann–Whitney test (non-normal distribution). Testing for normal distribution was performed with Shapiro-Wilk and Kolmogorov-Smirnow tests (GraphPad Prism V8.4.3. *p* values are presented as * *p* ≤ 0.05, ** *p* ≤ 0.01, *** *p* ≤ 0.001, and **** *p* ≤ 0.0001.

## 3. Results

### 3.1. Different Recruitment and Localization of Ly6C^hi^CD11b^+^ Monocytes in the Liver following Infection with E. histolytica or L. monocytogenes

Proinflammatory Ly6C^hi^ monocytes exhibit opposite functions in murine models for *E. histolytica* and *L. monocytogenes* liver infection. In the former, their absence resulted in the amelioration of parasite-dependent liver damage, whereas in the listeria model it exacerbated liver pathology [16,23]. To better understand the dynamics of monocyte recruitment, we examined hepatic protein concentrations of chemokines involved in these processes such as CCL2 and CCL3. We found a significant increase in CCL2 levels at d3, and an increase in CCL3 levels at d3 and d5 following parasitic infection (Figure 1A). During *L. monocytogenes* infection, CCL2 levels were higher and increased already at d1 post infection (p.i.), while CCL3 levels were lower than during parasitic infection or in naïve mice (Figure 1A). Additional cytokine analysis revealed significantly elevated expression of IL-1β, IL-10, and IL-13 during *E. histolytica* infection, as well as increased expression of TNF-α and IFN-γ during *L. monocytogenes* infection (Appendix A).

To determine monocyte recruitment, we isolated leukocytes from infected livers and measured the percentage of CD11b^+^Ly6C^hi^ and CD11b^+^Ly6C^lo^ monocytes by flow cytometry. At d3 p.i., the CD11b^+^Ly6C^hi^ monocyte populations increased in both infection models, whereas the CD11b^+^Ly6C^lo^ monocyte population increased only following *E. histolytica* infection (Figure 1B; absolute numbers see Appendix A). Staining of paraffin-embedded liver sections from both models (d3p.i.) with hematoxylin and eosin (H&E), anti-CD11b, and anti-Ly6C revealed accumulation of CD11b^+^ and Ly6C^+^ cells in a dense margin around the central amebic abscess (Figure 1C) while CD11b^+^ and Ly6C^+^ cells accumulated in the center of typical *L. monocytogenes*-induced granulomas (Figure 1C; controls see Appendix A).

In both infection models, the increase in CCL2 expression led to an increase in the proportion of Ly6C^hi^ monocytes in the liver, but with different localization in the affected tissue. The protective effect of monocytes from the listeria model already suggests heterogeneity of Ly6C^hi^ monocytes between the two infections.

### 3.2. Monocytes from Both Infection Models Show Significant Differences in Gene Expression

To gain a deeper understanding of the phenotype of the Ly6C^hi^ monocyte subset during hepatic amebiasis and listeriosis, 3 and 5 days p.i., CD11b^+^Ly6G^-^Ly6C^hi^ as well as Ly6C^lo^ monocytes were sorted from infected livers by flow cytometry. The RNA from both populations was extracted and subjected to RNA sequencing (Figure 2A).

Genes showing a significant difference in expression (adjusted *p* value < 0.05) in monocytes obtained from the two infection models at d3 p.i. were included in the analysis. 5486 genes were differentially expressed in Ly6C^hi^ monocytes from *E. histolytica*- and *L. monocytogenes*-infected mice, 54 genes were differentially expressed in Ly6C^lo^ cells from both models, and 194 genes were differentially expressed in both Ly6C^hi^ and Ly6C^lo^ monocytes in both infection models (Figure 2B). PANTHER GO analysis of genes differentially expressed in Ly6C^hi^ monocytes revealed that a small percentage of genes was included in the GO terms “immune systems process” (GO:0002376) and “response to stimulus” (GO:0050896) (Figure 2B). By contrast, we also observed the differential expression of genes associated with GO terms related to immune responses in anti-inflammatory Ly6C^lo^ monocytes (see Appendix A for a detailed list of highly regulated genes and GO terms).

Principal component analysis of Ly6C^hi^ and Ly6C^lo^ monocytes revealed clustering into different groups (Figure 2C). The day post infection did not affect the grouping, however there was a clear difference between Ly6C^hi^ monocytes from *L. monocytogenes*-infected mice (cluster A) and Ly6C^hi^ monocytes from *E. histolytica*–infected mice (cluster B): the latter clustered together with Ly6C^hi^ monocytes from naïve mice. Additional differences between Ly6C^hi^ and Ly6C^lo^ from the *E. histolytica* and the listeria model are also depicted by a volcano-plot and a heat map (see Appendix A). For example, differences include genes involved in proinflammatory IFN-γ related signaling (i.e., *Iigp1, Gbp2,* and *Gbp8*) in Ly6C^hi^ monocytes from *L. monocytogenes*-infected mice, but the upregulation of genes involved in anti-inflammatory, phagocytic, or metabolic processes (*Cx3cr1, Mfge8, Hpgd*) in Ly6C^hi^ monocytes from *E. histolytica*-infected mice. Overall, proinflammatory Ly6C^hi^ monocyte in both infection models differed significantly at the transcriptional level. Ly6C^hi^ monocytes from *L. monocytogenes* infected mice presented an upregulated expression of a large number of genes, including genes with a potential function in their antibacterial response. In contrast, changes in the gene expression of Ly6C^hi^ monocytes from *E. histolytica* infected mice were less pronounced and a large part of their expression profile was shared with Ly6C^hi^ monocytes from naïve mice.

### 3.3. Ly6C^hi^ Monocytes from L. monocytogenes-Infected Mice Have an Activated Phenotype and Lower M2 Polarization at the Transcriptional Level Than Ly6C^hi^ Monocytes from E. histolytica-Infected Mice

By focusing on transcription factors with putative relevance to the polarization of monocytes towards classically activated M1 or alternatively activated M2 macrophages, we found that during infection with *L. monocytogenes*, the activation and development of proinflammatory monocytes is characterized by factors, such as *Irf1, Irf2, Ifi204, Batf2, Mndal,* and *Irf7* [24,25,26,27,28] (Figure 3A). By contrast, during *E. histolytica* infection, Ly6C^hi^ monocytes are characterized by the upregulation of transcription factors *Mafb, Hes1, Fos,* and *Tsc22d3* (Figure 3A), which contribute to an anti-inflammatory and regenerative phenotype [29,30,31,32,33]. Time-course analysis of the expression data shows that, in addition to other genes, a selection of the above factors exhibits significantly different expression patterns between the two infection models, starting as early as d3 p.i. and remaining different until d5 p.i. (Figure 3B).

Fate mapping and transfer approaches have revealed the marked plasticity of proinflammatory monocytes and identified distinct routes of monocyte polarization. Such studies suggest the existence of novel monocyte subsets, such as “Ly6C^hi^ to Ly6C^lo^”-converting monocytes, nMO, dcMO, and *Cxcl10*^+^ and *Saa3*^+^ monocytes [12,13,14]. According to this classification, the Ly6C^hi^ monocytes triggered by *E. histolytica* infection would appear to belong to the Ly6C^hi^ to Ly6C^lo^-converting monocyte subset. By contrast, with the exception of *Csf1* and *Ly6c2 (Ly6C)*, the respective genes were downregulated in monocytes from the *L. monocytogenes* infection model (Figure 3C) [35]. Relevant genes related to nMO development were upregulated in monocytes from *L. monocytogenes*-infected mice (Figure 3C). When we considered the genes that define dcMO, we found an intermediate picture. MHC-II related genes were upregulated in monocytes from *L. monocytogenes*-infected mice. However, some other hallmark genes of dcMO (i.e., *Flt3, Pid1*, and *Hpgd*) were strongly downregulated. Moreover, with the exception of *il1b,* signature genes of *Cxcl10*^+^ and *Saa3*^+^ monocytes were also upregulated in Ly6C^hi^ monocytes from *L. monocytogenes*-infected mice, indicating a broad repertoire of putative new Ly6C^hi^ monocyte subsets during this type of infection (Figure 3C). However, several other relevant genes involved in proinflammatory or anti-inflammatory immune processes are additionally upregulated in monocytes following *E. histolytica* infection. Among these are *Cd14*, *Trem2*, a negative immune regulator and marker for M2 polarization [36], as well as *Arg1* and *Arg2*, further indicating the transition from pro-to anti-inflammatory monocytes (Figure 3D).

In summary, we were able to assign monocytes from both infection models to recently suggested subgroups with a more proinflammatory and activated phenotype in the *L. monocytogenes* model (nMO; dcMO; *Cxcl10*^+^ and *Saa3^+^* monocytes) and a scarcely activated phenotype in the parasite model characterizing Ly6C^hi^ to Ly6C^lo^ converting monocytes.

### 3.4. Surface Marker Expression Implies Pathogen-Dependent Subsets of Proinflammatory Monocytes

Next, we analyzed the differential expression of genes encoding surface markers that may be useful for further subdivision of Ly6C^hi^ monocytes. We found significant upregulation of genes encoding *Ly6c2, Cd38,* and *Cd74* in Ly6C^hi^ monocytes from *L. monocytogenes*-infected mice, whereas higher expression of *Cd14* was characteristic for Ly6C^hi^ monocytes from *E. histolytica*-infected mice (Figure 4A,B). The expression of these genes at the protein level on Ly6C^hi^ monocytes from both infection models was validated by flow cytometry. Although not significantly regulated at the transcriptional level, we included the analysis of the co-stimulatory receptor CD86 in the panel to further describe M1 polarization of inflammatory monocyte [37]. As in monocytes from naïve mice and in agreement with the transcriptomic data, we found that the percentage of CD14-expressing Ly6C^hi^ monocytes was higher and remained higher from d1 p.i. on following *E. histolytica* infection than in Ly6C^hi^ monocytes from *L. monocytogenes*-infected mice, the latter initially decreased but increased from d5 of infection (Figure 4C). This picture was mirrored by lower MFIs for CD14 on Ly6C^hi^ monocytes derived from the *L. monocytogenes* model than on monocytes from the *E. histolytica* model (Figure 4D). When compared with that in uninfected animals, expression of CD38 increased significantly in both models shortly after infection. However, expression was significantly stronger on monocytes from the *L. monocytogenes* infection model (Figure 4D). Initially, the percentage of CD74-expressing monocytes remained the same as that in naïve mice following *E. histolytica* infection, but decreased on d5 p.i. (Figure 4C). During *L. monocytogenes* infection, expression and MFI of CD74 increased on d1 p.i., but then decreased to the level observed in uninfected animals as infection progressed (Figure 4C,D). The expression and MFI level of CD86 also decreased significantly over time during infection with *L. monocytogenes*, and to a lesser extent this was also true for monocytes in the *E. histolytica* model (Figure 4C) (expression on monocytes from spleen, blood, bone marrow see Appendix A).

Overall, the results of the transcriptome analysis of surface marker expression, with the exception of CD74, are reflected at the protein level in vivo. Furthermore, they suggest that CD14 in combination with CD38 may be additional putative markers for a Ly6C^hi^ monocyte subset that is very different from the conventional proinflammatory Ly6C^hi^ monocyte subset.

To further differentiate Ly6C^hi^ monocytes, we selected molecules that were shown by transcriptome analysis to be highly expressed by Ly6C^hi^ monocytes after *E. histolytica* infection (*Mafb*) or *L. monocytogenes* infection (*Irf2*) (Figure 3B). However, MAFB1 protein was excluded from further analysis since less than 1% of Ly6C^hi^ monocytes expressed the protein (data not shown). As seen for mRNA, protein expression of IRF2 was significantly stronger in Ly6C^hi^ monocytes after *L. monocytogenes* infection than in monocytes from naïve or *E. histolytica*-infected mice (Figure 5A). On day 3 after infection, when the most severe symptoms in both models appear, the combination of antibodies against IRF2 and CD14 resulted in the detection of a significantly increased monocyte population after *E. histolytica* infection. When IRF2 detection was combined with the detection of CD38 and CD86, Ly6C^hi^ monocytes after infection with *L. monocytogenes* were significantly different from those of naive or *E. histolytica*-infected animals, whereas the combination of IRF2 detection with CD74 did not reveal significantly different monocyte subpopulations between the two infection models (Figure 5B).

After analysis of the distribution of Ly6C^hi^ monocyte subpopulations by UMAP, initially considering only surface markers, clear demarcation of populations by *L. monocytogenes* and *E. histolytica* infection and by naïve animals was seen, with overlapping populations of the latter (Figure 5C). When Irf2 was included, 6 distinct clusters were identified (Figure 5D). A distinct IRF2-positive monocyte population and a greater heterogeneity in the area of monocytes from the listeria model and a clear delineation of monocyte populations from naive and *E. histolytica*-infected animals could be visualized in the UMAP analysis after clustering with FlowSOM and Cluster Explorer analysis (Figure 5D).

In summary, we found that the combination of antibodies against IRF2 with antibodies against CD14, CD38, or CD86 helped to distinguish proinflammatory Ly6C^hi^ monocytes from both infection models, supporting the results of the transcriptome study (Figure 3C) in that these monocytes are already in a transitional stage to anti-inflammatory monocytes.

### 3.5. CD38^+^Ly6C^hi^ Monocytes Produce ROS and Contribute to Monocyte-Dependent Immunopathology during Hepatic Amebiasis

In addition to the production of proinflammatory cytokines and chemical mediators, activated Ly6C^hi^ monocytes also express ROS [38]. Based on the higher mRNA expression of genes involved in ROS production and NADPH oxidase (i.e., *Sod**1*, *Sod2*, *Ncf1*, *Ncf**4* as well as *Nox2 and Noxred 1*) by Ly6C^hi^ monocytes from *L. monocytogenes*-infected compared with *E. histolytica*-infected mice (Figure 6A), we analyzed ROS production by Ly6C^hi^ monocytes in both models (Figure 6B). Consistent with the transcriptomic results, and consistent with a low MFI, the percentage of ROS^+^Ly6C^hi^ monocytes decreased during *E. histolytica* infection (Figure 6C) but increased during *L. monocytogenes* infection (Figure 6D).

To further characterize ROS^+^ Ly6C^hi^ monocytes, we examined the expression of surface marker CD38, which exhibits various functions during cell activation [39]. Interestingly, although the number of ROS-producing Ly6C^hi^ monocytes decreased during infection with *E. histolytica*, the percentage of ROS^+^ CD38^+^ out of Ly6C^hi^ monocytes increased rapidly, and remained elevated, during infection (Figure 6E).

As expected, the proportion of these cells also increased during infection with *L. monocytogenes*, but with a delay compared with parasitic infection, and the final proportion was higher (Figure 6E). Next, we used knockout mice lacking CD38 [40,41] to investigate whether CD38^+^Ly6C^hi^ monocytes contributes to abscess formation during *E. histolytica* liver infection. *Cd38^-/-^* mice had significantly smaller abscesses on d3 p.i. (Figure 6F) and a significantly lower level of proinflammatory monocytes (similar to naïve mice) (Figure 6G).

In summary, ROS-production as well as the expression of CD38 characterizes the true proinflammatory phenotype of Ly6C^hi^ monocytes in both infection models.

## 4. Discussion

Monocytes are critical for the defense against microbial infections, but also for promoting resolution of inflammation. However, an improper balance of these tasks can lead to the collateral damage of host tissues and delay of tissue regeneration [42].

The rationale for the present study arose from the striking differences in the function of classical proinflammatory Ly6C^hi^ monocytes revealed by *Ccr2*^-/-^. Whereas the lack in the egress from the bone marrow and hence the recruitment of Ly6C^hi^ monocyte prevented liver destruction after *E. histolytica* infection [16], their absence in *L. monocytogenes* infection exacerbated disease progression [23], pointing to functional differences in this monocyte subset which was originally regarded as homogeneous. Recent studies based on single-cell sequencing actually suggest an even greater diversity. Under conditions of homeostasis three more proinflammatory monocyte subsets: nMO, dcMO [12,13,43,44], and *Cxcl10*^+^ and *Saa3*^+^ monocytes were identified [14]. Interestingly, all analyzed genes involved in development of nMO were strongly upregulated in proinflammatory monocytes during *L. monocytogenes* infection, but they were unaffected in monocytes from the amebic model. Enhanced development of Ly6C^hi^ monocytes into nMO has been demonstrated previously, but only under LPS stimulation [45] and the present study is the first to demonstrate its presence in vivo by bacterial infection. Likewise, genes related to MHC-II-mediated antigen presentation and dcMO development were upregulated in listeria infection. However, some factors thought to be important for dcMO development (i.e., *Flt3*, *Pou2f2*, *Pid1*, and *Hpgd*) were strongly downregulated while their expression by monocytes from the amebiasis model was comparably higher. In addition, genes characteristic for the subset of *Cxcl10*^+^ and *Saa3*^+^ monocytes that arise under sterile inflammatory conditions (e.g., autoimmune encephalitis) [14] were only upregulated in the listeria model. Taken together, the data suggest that proinflammatory monocytes from *L. monocytogenes*-infected animals display a distinct proinflammatory phenotype, characterized by the upregulation of genes associated with nMO, dcMO and *Cxcl10*^+^ and *Saa3*^+^ cells.

Additional relevant transcriptional differences between Ly6C^hi^ monocytes from both models became apparent by examining the expression of selected transcription factors and genes involved in polarization of monocytes. Ly6C^hi^ monocytes from the parasite model exhibited a more anti-inflammatory phenotype and the expression of genes involved in conversion of Ly6C^hi^ monocytes to Ly6C^lo^ monocytes. Altered genes include *Nr4a1*, a major transcription factor responsible for transition of Ly6C^hi^ to Ly6C^lo^ and survival of Ly6C^lo^ cells [30,46], while *Csf1* and *Ly6c2,* promoting survival and activation of Ly6C^hi^ monocytes, were downregulated [35,47]. Their further polarization towards anti-inflammatory M2 macrophages [9] is supported by the upregulation of *MafB, Nr4a1*, or *Fos* [29,30,33]. As already evident from the cluster analysis, Ly6C^hi^ monocytes from the parasite model were overall quite similar to those from naïve animals. However, some genes were differentially regulated, e.g., *Arg1/Arg 2*, further suggesting an ongoing polarization into an anti-inflammatory M2 phenotype. In contrast, monocytes from the *L. monocytogenes* infection model were characterized by a classical proinflammatory, interferon-driven transcription factor-like profile (i.e., *Mndal*, *Ifi204* and *Irf2*) [25,28], pointing towards an M1 phenotype [9].

A suitable antibody panel to distinguish bona fide inflammatory Ly6C^hi^ monocytes from non-inflammatory Ly6C^hi^ monocytes and to study their dynamics of Ly6C^hi^ monocytes in both infection models, was developed by selecting several surface markers that had emerged from transcriptome analysis. These included activation markers such as CD38 [39,40,41], CD74, a receptor for proinflammatory macrophage migration inhibitory factor involved in cell proliferation and antigen presentation [48,49] as well as CD14, co-receptor for several Toll-like receptors involved in proinflammatory processes [50]. While CD38 and CD74 were more highly expressed in Ly6C^hi^ monocytes from the *L. monocytogenes* infection model, CD14 was the only surface marker with higher expression in Ly6C^hi^ monocytes from the parasite model. Although not differentially expressed on the mRNA level, we included CD86 as an additional proinflammatory M1 marker [41].

The expression of CD14 and CD74 on Ly6C^hi^ monocytes from the parasite model was very similar to those from naïve mice, whereas CD14 in the *L. monocytogenes* model initially decreased and only increased toward the end of the disease course. CD38 was more highly expressed on monocytes from the parasitic model during the early phase of infection, thus describing inflammatory Ly6C^hi^ before transitioning to Ly6C^lo^ cells. As expected, CD38, CD74, and to a lesser extent CD86 remain highly expressed in Ly6C^hi^ monocytes from the *L. monocytogenes* model over time, suggesting that these markers are useful for further distinguishing proinflammatory Ly6C^hi^ monocytes. Next, we included the transcription factor IRF2 within the panel. IRF2 in combination with CD14 best distinguishes the Ly6C^hi^ population in the ameba and listeria model from the population in naïve mice at least on day 3 post infection, the peak of liver pathology in these models (Figure 5B). Subsequent UMAP analysis based on the designated surface markers confirmed clear delineation of Ly6C^hi^ monocyte populations between naïve, *E. histolytica*- and *L. monocytogenes*-infected mice, as well as the highest diversity of proinflammatory monocyte subpopulations from the *L. monocytogenes* model. Finally, we used ROS production as a hallmark of proinflammatory monocyte activation [38]. In contrast to the *E. histolytica* model, where it remained stable, we observed a continuous increase in ROS-producing Ly6C^hi^ monocytes expressing CD38 in the listeria model. Up-regulation of CD38 on monocytes during infection with listeria has been described previously [39]. Interestingly, genetic deletion of CD38 resulted in increased accumulation of inflammatory monocytes in the liver but not in the spleen and was associated with higher susceptibility to listeria infection, as observed during genetic deletion of *Ccr*2 [23,39]. Assuming that CD38^+^ROS^+^ monocytes are responsible for immunopathological mechanisms during the early phase of hepatic amebiasis, genetic deletion of CD38 should lead to smaller abscesses. Indeed, we were able to detect this phenotype, and it was associated with a marked reduction in the proportion of recruited Ly6C^hi^ monocytes.

In summary, analysis of Ly6C^hi^ monocyte populations from two different infection models shows that proinflammatory Ly6C^hi^ monocytes differ depending on the infectious agent. Based on the present results, we propose that the addition of IRF2, CD14, and CD38 or CD86 to the classical markers (CD11b, Ly6C, Ly6G) can help distinguish true proinflammatory Ly6C^hi^ monocytes from non-inflammatory Ly6C^hi^ monocytes.

## Figures and Tables

**Figure 1 cells-11-02539-f001:**
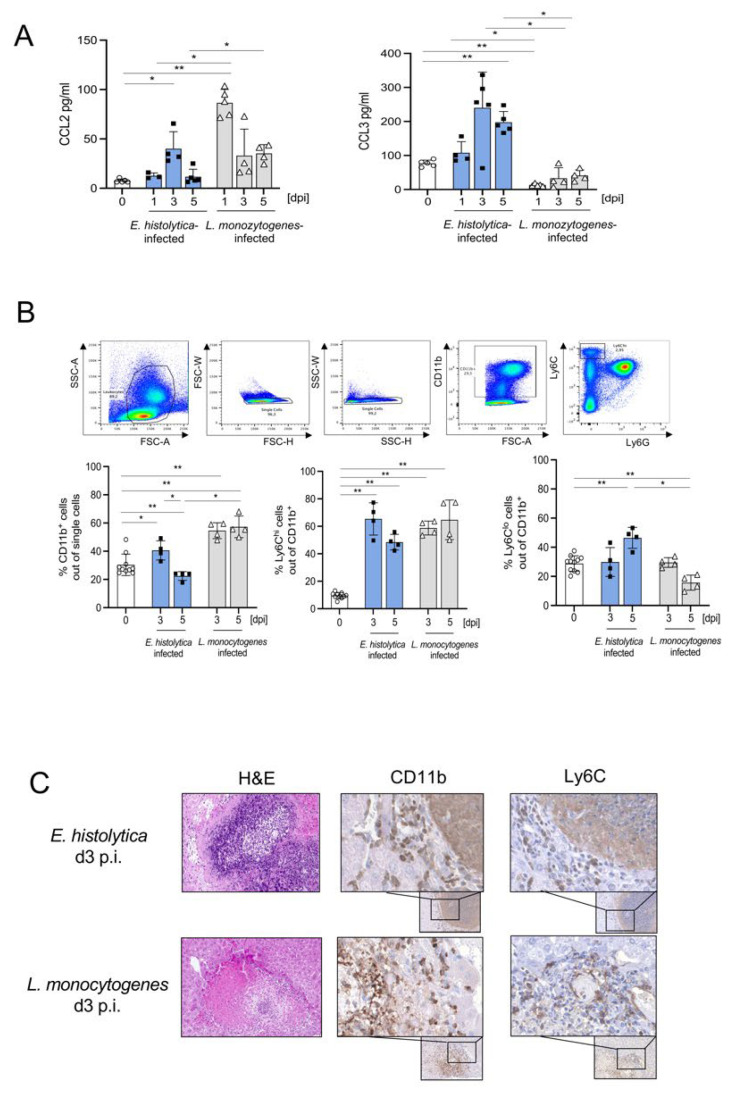
Recruitment of CD11b^+^Ly6C^hi^ and Ly6C^lo^ monocytes to the liver during infection with *E. histolytica* and *L. monocytogenes*, and protective properties of Ly6C^hi^ monocytes against *E. histolytica* infection. (**A**) Expression of CCL2 and CCL3 in liver lysates after intrahepatic *E. histolytica* (2 × 10^5^) and systemic *L. monocytogenes* (2 × 10^4^) infection, measured in a multiplex cytokine assay. (**B**) Gating scheme and percentages of CD11b^+^, CD11b^+^ Ly6C^hi^, and Ly6C^lo^ monocytes in the liver following *E. histolytica* or *L. monocytogenes* infection. (**C**) Liver sections from *E. histolytica*- and *L. monocytogenes*-infected mice were stained with H&E, anti-CD11b (EPR1344), and anti-Ly6C (ER-MP20) on day 3 p.i. (**A**–**C**) One representative experiment of three is shown. (* *p* < 0.05; ** *p* < 0.01; Mann-Whitney U test).

**Figure 2 cells-11-02539-f002:**
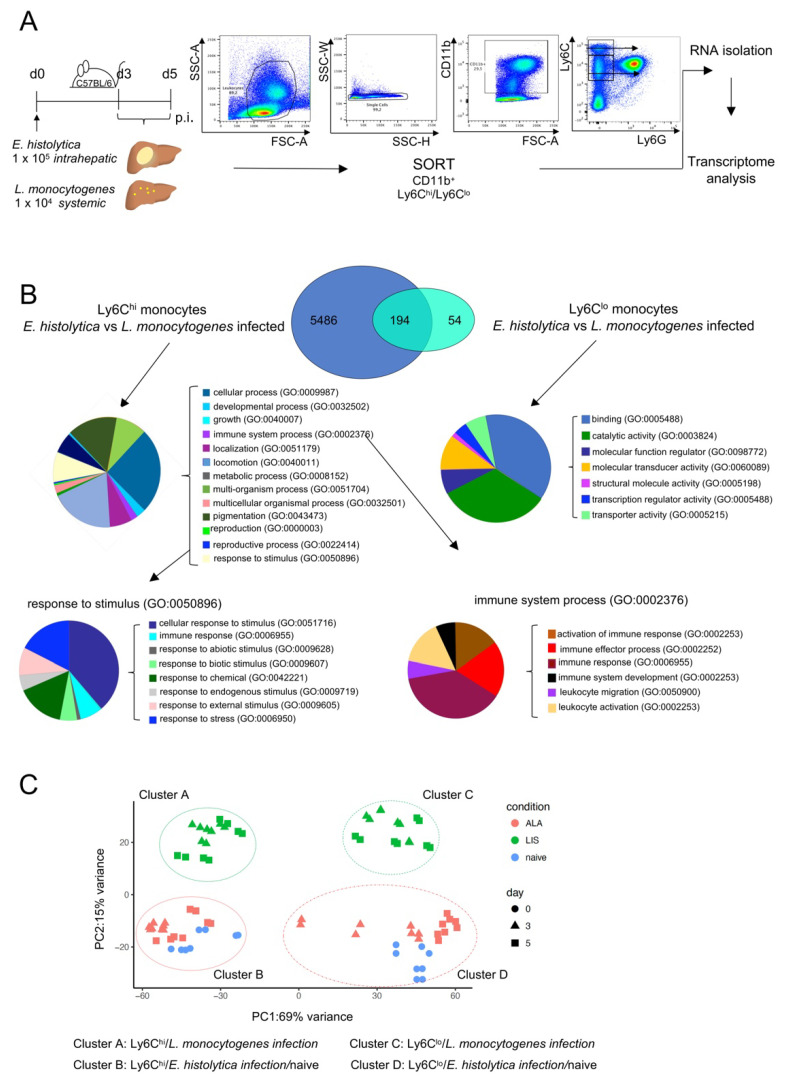
Transcriptome analysis of monocytes from the livers of mice infected with E. histolytica or L. monocytogenes. (**A**) Gating and mRNA purification strategy for liver-specific Ly6C^hi^ and Ly6C^lo^ monocytes from *E. histolytica-* and *L. monocytogenes*-infected mice. (**B**) VENN diagram of transcriptome analysis of Ly6C^hi^ and Ly6C^lo^ monocytes shows significantly regulated genes of Ly6C^hi^ and Ly6C^lo^ monocytes from both infection models. PANTHER GO-slim analysis was performed to identify biological pathways involving significantly regulated genes in both monocyte populations. (**C**) Principal component analysis of transcriptomic data from Ly6C^hi^ and Ly6C^lo^ monocytes from naïve and *E. histolytica-* and *L. monocytogenes*-infected mice.

**Figure 3 cells-11-02539-f003:**
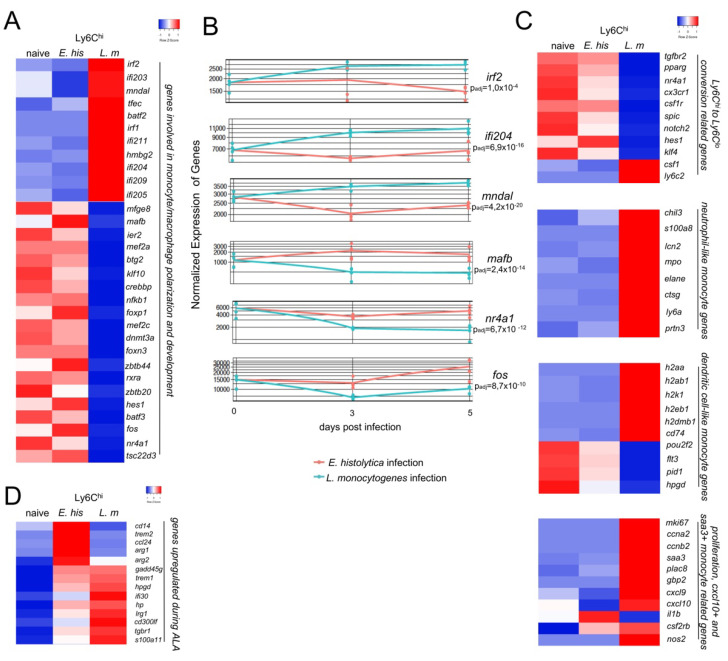
Early M2 polarization and a less activated mRNA expression profile in Ly6C^hi^ monocytes from *E. histolytica*-infected compared to *L. monocytogenes*-infected mice. (**A**) Heat map showing selected regulated genes (p_adj_ < 0.05; foldchange > 2) involved in monocyte/macrophage polarization and activation of Ly6C^hi^ monocytes derived from the livers of *E. histolytica* (*E. his*)- and *L. monocytogenes* (L. m)-infected. (**B**) Time-course analysis of mRNA encoding M2 transcription factors and interferon-regulated/activated factors. (**C**) Heatmap showing classification of Ly6C^hi^ monocytes derived from the livers of both infection models according to expression of genes involved in conversion from Ly6C^hi^ to Ly6C^lo^, neutrophil-like, dendritic cell-like, or *Cxcl10*^+^ and *Saa3*^+^-like monocytes. (**D**) Heat map of selected genes upregulated during ALA. All heatmaps were designed using the online tool “heatmapper” [34].

**Figure 4 cells-11-02539-f004:**
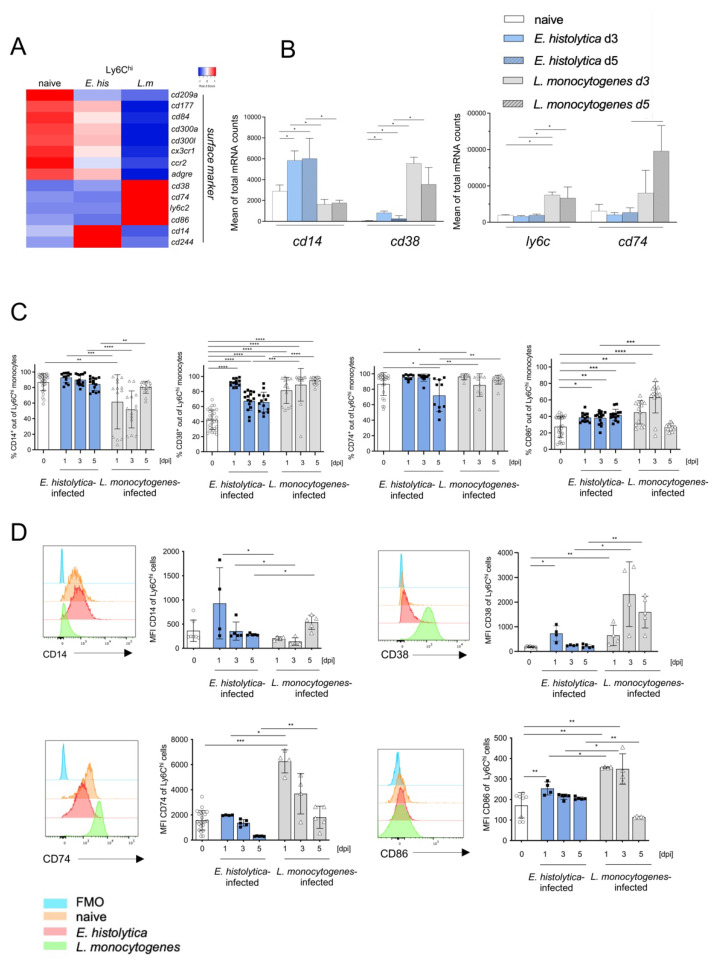
Marked differences in expression of surface markers by Ly6C^hi^ monocytes after infection with *E. histolytica* or *L. monocytogenes* (**A**) Heatmap depicting differential expression of mRNA encoding selected markers on the surface of liver Ly6C^hi^ monocytes after infection with *E. histolytica* (*E. his*) or *L. monocytogenes* (*L. m*). (**B**) Normalized mRNA counts of selected surface marker genes (from transcriptome analysis). (**C**) Percentage of CD14^+^, CD38^+^, CD74^+^, and CD86^+^ Ly6C^hi^ monocytes during the course of infection at the indicated time points post-infection (measured by flow cytometry). (**D**) Histogram and MFI of CD14^+^, CD38^+^, CD74^+^, and CD86^+^ Ly6C^hi^ monocytes during the course of infection. Data in C were pooled from three independent experiments. Data in D are representative of one of these three experiments and all data are presented as the mean ± SEM (* *p* < 0.05; ** *p* < 0.01; *** *p* < 0.001, **** *p* < 0.0001; Mann-Whitney U test).

**Figure 5 cells-11-02539-f005:**
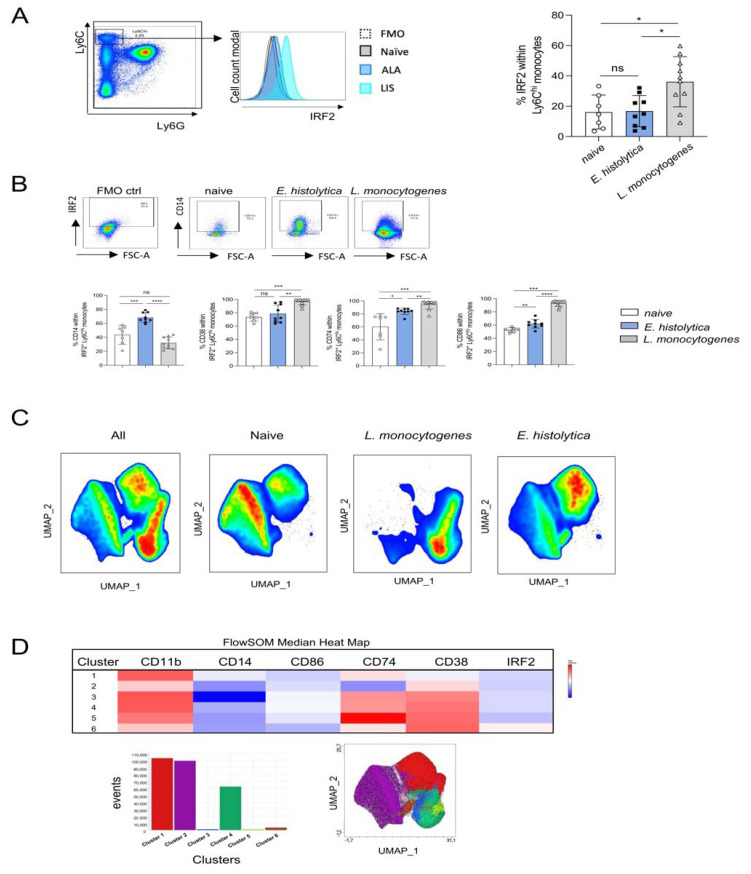
New Ly6C^hi^ monocyte subsets identified according to expression of selected surface markers and IRF2. (**A**) Gating strategy, histogram, and percentage of IRF2^+^ Ly6C^hi^ monocytes from livers of naïve mice and from *E. histolytica*- and *L. monocytogenes*-infected mice on day 3 post infecion. (**B**) Gating strategy (exemplary for CD14^+^ gated cells) based on the FMO control to determine CD14^+^, CD38^+^, CD74^+^, and CD86^+^ monocytes within the IRF2^+^ Ly6C^hi^ monocyte population from naïve mice and from both infection models on day 3 post infection. (**C**) UMAP plots of Ly6C^hi^ monocytes (including CD14^+^, CD86^+^, CD74^+^, CD38^+^) from naïve mice and from *E. histolytica*- and *L. monocytogenes*-infected mice (down sampled to 90,000 cells per sample). Six samples per source material were included in the analysis. (**D**) Cluster heatmap table showing surface marker and IRF2 expression by Ly6C^hi^ monocytes, cluster events, and integration of cluster events in the UMAP plot (* *p* < 0.05; ** *p* < 0.01; *** *p* < 0.001; **** *p* < 0.0001; Mann-Whitney U test).

**Figure 6 cells-11-02539-f006:**
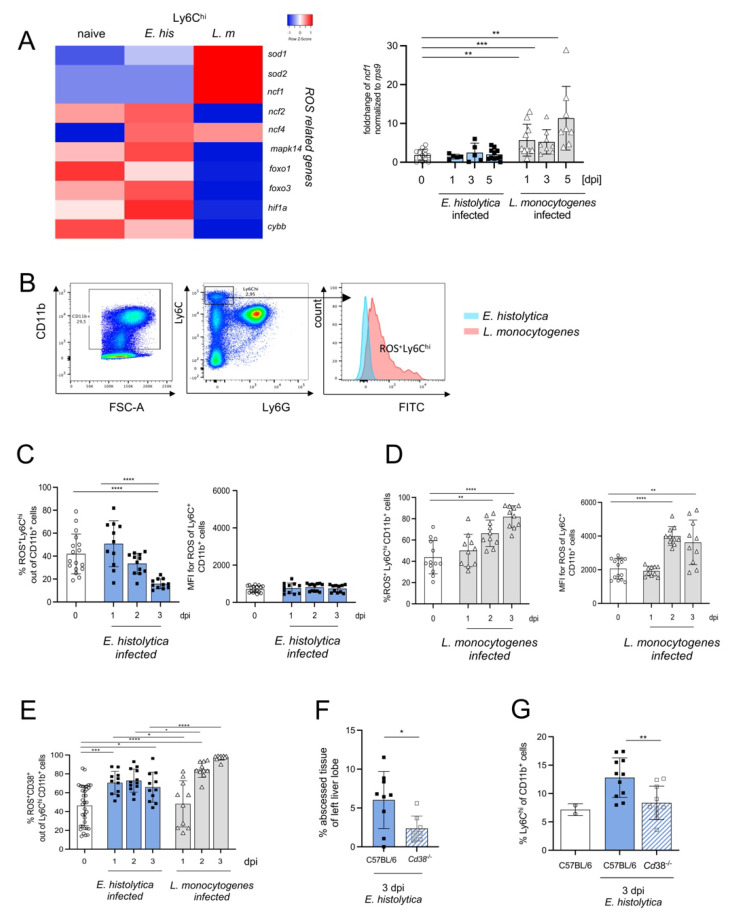
CD38^+^Ly6C^hi^ monocytes represent the major monocytic source of ROS production and contribute to liver damage following *E. histolytica* infection. (**A**) Heat map of differentially regulated genes involved in ROS production and a graph showing fold changes in expression of *Ncf1* mRNA in the liver during infection with *E. histolytica* or *L. monocytogenes*. (**B**) Gating scheme and histogram for ROS^+^ liver Ly6C^hi^ monocytes. Percentage and mean fluorescent intensity (MFI) of ROS^+^Ly6C^hi^ monocytes during (**C**) *E. histolytica* and (**D**) *L. monocytogenes* infection. (**E**) Percentage of ROS^+^CD38^+^ out of Ly6C^hi^CD11b^+^ monocytes in both infection models. (**F**) Percentage of amebic liver abscess weight in relation to the left liver lobe in WT (C57BL/6) and *Cd38*^-/-^ mice. (**G**) Percentage of Ly6C^hi^ CD11b^+^ cells in the liver in naive, infected WT and *Cd38*^-/-^ mice. (* *p* < 0.05; ** *p* < 0.01; *** *p* < 0.001, **** *p* < 0.0001; Mann-Whitney U test).

## Data Availability

Not applicable.

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
