# Peer review of "The Properties of Proinflammatory Ly6C^hi^ Monocytes Are Differentially Shaped by Parasitic and Bacterial Liver Infections"

_cells, 2022, doi:10.3390/cells11162539_

Round 1
Reviewer 1 Report
General comments:
Recent progress and ongoing monocyte research have modified our ability to understand their impact on immune response and inflammation. Therefore, the subject of the present study is interesting and up-to-date.
Specific comments:
This is an elegant and comprehensive investigation of Ly6Chi monocytes in 2 different models of infection i.e. parasitic and bacterial. The authors indicate that based on new markers that were identified in the study, the aforementioned monocytes can be further subclassified into different subpopulations.
The manuscript is written in an easy-to-understand way. It is relevant for the immunology field and presented in a well-structured manner. The experimental design is appropriate to test the hypothesis.
The conclusions are consistent with the evidence and arguments presented.
I would only suggest deleting the repeated word "monocytes" in line 447.
Conclusion: I recommend accepting the manuscript in its present form.
Author Response
Dear reviewer, thank you very much for the positive and encouraging review. We have deleted the repetition of the word "monocytes" not in lane 447, but in lane 413 and hope that this was in your sense.
With best regards
Hanna Lotter
Reviewer 2 Report
In the manuscript, using two mouse models of Entamoeba histolytica (ameba) and Listeria monocytogenes liver infections, the authors analyzed differences in Ly6Chi monocytes subsets from the liver. They revealed that monocytes from both infection models show significant differences in gene expression; the recruited Ly6Chi monocytes in the ameba infection model had elevated expression of M2-related transcription factors (Mafb, Nr4a1, Fos, etc.), whereas Ly6Chi monocytes in the listeria infection model expressed M1-related transcription factors (Irf2, the Mndal, Ifi204, etc.). In other words, infection with the ameba induced a significantly lower inflammatory phenotype in Ly6Chi monocytes compared to infection with the listeria.
Also, the levels of CD38 and CD74 were higher expressed in Ly6Chi monocytes from the listeria infection model, whereas the level of CD14 was the only surface marker with higher expression in Ly6Chi monocytes from the ameba infection model. Furthermore, contrary to the ameba infection model, where ROS production remained stable, they observed a continuous increase in ROS-producing Ly6Chi monocytes expressing CD38 in the listeria model. The authors used CD38-knockout mice and found that genetic deletion of CD38 resulted in increased accumulation of inflammatory monocytes in the liver and was associated with higher susceptibility to listeria infection.
The manuscript is clearly written and well supported by extensive and rigorous data sets. Therefore, I do not have any substantial amendments to suggest.
Author Response
Dear reviewer, thank you very much for the positive and encouraging review.
With best regards,
Hanna Lotter